# Soybean–SCN Battle: Novel Insight into Soybean’s Defense Strategies against *Heterodera glycines*

**DOI:** 10.3390/ijms242216232

**Published:** 2023-11-12

**Authors:** Sepideh Torabi, Soren Seifi, Jennifer Geddes-McAlister, Albert Tenuta, Owen Wally, Davoud Torkamaneh, Milad Eskandari

**Affiliations:** 1Department of Plant Agriculture, University of Guelph, Guelph, ON N1G 2W1, Canada; storabi@uoguelph.ca; 2Aurora Cannabis Inc., Comox, BC V9M 4A1, Canada; soren.seifi@gmail.com; 3Department of Molecular and Cellular Biology, University of Guelph, Guelph, ON N1G 2W1, Canada; jgeddesm@uoguelph.ca; 4Ontario Ministry of Agriculture, Food and Rural Affairs, Ridgetown, ON N0P 2C0, Canada; albert.tenuta@ontario.ca; 5Harrow Research and Development Centre, Agriculture and Agri-Food Canada, London, ON N0R 1G0, Canada; owen.wally@agr.gc.ca; 6Département de Phytologie, Université Laval, Québec City, QC G1V 0A6, Canada; davoud.torkamaneh.1@ulaval.ca

**Keywords:** soybean, soybean cyst nematode (SCN), dual-RNA sequencing, host and pathogen interaction

## Abstract

Soybean cyst nematode (SCN, *Heterodera glycines*, Ichinohe) poses a significant threat to global soybean production, necessitating a comprehensive understanding of soybean plants’ response to SCN to ensure effective management practices. In this study, we conducted dual RNA-seq analysis on SCN-resistant Plant Introduction (PI) 437654, 548402, and 88788 as well as a susceptible line (Lee 74) under exposure to SCN HG type 1.2.5.7. We aimed to elucidate resistant mechanisms in soybean and identify SCN virulence genes contributing to resistance breakdown. Transcriptomic and pathway analyses identified the phenylpropanoid, MAPK signaling, plant hormone signal transduction, and secondary metabolite pathways as key players in resistance mechanisms. Notably, PI 437654 exhibited complete resistance and displayed distinctive gene expression related to cell wall strengthening, oxidative enzymes, ROS scavengers, and Ca^2+^ sensors governing salicylic acid biosynthesis. Additionally, host studies with varying immunity levels and a susceptible line shed light on SCN pathogenesis and its modulation of virulence genes to evade host immunity. These novel findings provide insights into the molecular mechanisms underlying soybean–SCN interactions and offer potential targets for nematode disease management.

## 1. Introduction

Nematodes are multicellular animals and belong to the superphylum Ecdysozoa. Nematodes are parasitic and free-living worms that can shed their cuticle to grow [1]. Plant-parasitic nematodes are recognized as major agricultural pathogens [2]. One of the most destructive plant-parasitic nematodes that can result in significant yield losses and economic damage is soybean cyst nematode (SCN), caused by *Heterodera glycines* (HG) Ichinohe [3]. SCN poses a serious threat to world soybean production, causing annual yield losses in excess of USD 1.5 billion in North America [4].

Soybeans, as with other plants, fight off pathogens through the initiation of an array of defense mechanisms. A plant defense system is a complex system that consists of several lines of defense. The first layer is the plant’s passive defense mechanism. These include structural barriers, such as the cell wall that can physically block the entry of pathogens into the plant tissues [5,6]. If pathogens are successful in passing a plant’s passive defense mechanism, they have the chance to access the nutrients from the plant. However, after the passive defense mechanism, plants still possess two layers of actively induced immune systems. The first layer of the immune response, which is activated by a pathogen-associated molecular pattern (PAMP), is PAMP-triggered immunity (PTI). PAMPs are a diverse set of microbial molecules that have conserved structures perceived by plant surface-exposed receptors called pattern recognition receptors (PRRs) [7,8]. These membrane-bound PRRs are receptor-like kinases (RLK) or receptor-like proteins (RLP) with a high variety of intracellular domains. After the perception of PAMPs through PRRs, PTI will be activated as the first layer of the plant immunity system or surface immunity, which restricts pathogen proliferation. PTI signaling components are often targeted by various pathogen virulence effector proteins, resulting in diminished plant defenses and increased pathogen virulence. While PAMPs are conserved molecules that are shared among many different pathogens, effectors are typically species-, race-, or strain-specific molecules that contribute to pathogen virulence by targeting specific host plant processes [9]. Some plant resistance (R) proteins have evolved to recognize pathogen effectors directly or indirectly by associating with cytoplasmic immune receptors [10]. These receptors, which often contain a nucleotide-binding leucine-rich repeat domain, activate the second layer of immunity known as effector-triggered immunity (ETI) [8]. In contrast to PTI, ETI is a highly specific defense response that is triggered by the recognition of pathogen effectors that have been specifically adapted to interact with and manipulate host proteins. It can trigger programmed cell death, called the hypersensitive response (HR), in response to a pathogen attack, which helps to contain the infection and limit the damage caused to the plant [11].

The most effective management strategy to control SCN populations is using resistant cultivars rather than any other strategies, such as crop rotation or nematicides. Three popular Plant Introductions (PIs) resistant to SCN are PI 437654, PI 548402 (also known as Peking), and PI 88788, which carry resistant loci effective against multiple nematode races [12,13]. However, among all these resistant lines, plant breeders have heavily overused PI 88788 in the past three decades as the resistant parent in breeding programs. This has led to the selection of virulent biotypes of SCN and population shifting such as HG type 1.2.5.7, which are able to overcome the PI 88788-type resistance.

Soybean cyst resistance is a complex trait with polygenic inheritance. The first quantitative trait loci (QTL) underlying the resistance to *H. glycines* (i.e., rhg) were reported in the early 1960s [14,15]. Among several reported QTL, the QTL on chromosomes 18 (*rhg1*) and 8 (*Rhg4*) are the two major resistance ones that have been consistently mapped and reported in a variety of soybean germplasm [13,16,17]. In some SCN-resistant lines, such as PI 88788, *rhg1* with recessive action is sufficient to provide resistance to certain races of SCN and display an incompatible interaction with the nematode [13]. In other resistant sources, such as PI 548402, resistance to SCN requires both *rhg1* and *Rhg4*, while *Rhg4* exhibits a dominant gene action [18]. Brucker et al. 2005 [12] classified *rhg1* into two types: *rhg1-a* in PI 548402 (also known as Peking-type) with low copy number (three or fewer repeats), which reacts with *Rhg4* to provide greater resistance to SCN, and *rhg1-b* in PI 88788-type soybeans, which poses high copy number (four or more repeats) and provides the resistance without interacting with *Rhg4* [12,19]. SCN-susceptible lines such as Williams 82 and Lee 74 have only a single copy of *rhg1* [19]. Previous studies discovered that the SCN resistance governed by *rhg1* is mediated through a 31-kb segment that is tandemly repeated and carries three genes, including a predicted amino acid transporter (*Glyma18g02580*), an SNAP protein predicted to participate in the disassembly of SNARE membrane trafficking complexes (*Glyma18g02590*), and a protein with aWI12 (wound-inducible protein 12) region without functionally characterized domains (*Glyma18g02610*) [19,20]. On the other hand, map-based cloning of the *Rhg4* locus revealed that a single gene encoding a serine hydroxymethyltransferase (SHMT, *Glyma08g11490*) is responsible for the resistance [21].

By deploying defense mechanisms, SCN-resistant soybean genotypes are able to mount effective immune responses against SCN and minimize the damage caused to their production. However, pathogens have also evolved sophisticated strategies to evade or overcome these defenses, which has led to an ongoing evolutionary arms race between plants and pathogens. Therefore, studying soybean–SCN interactions can be challenging due to the complex nature of these interactions and the fact that the molecular mechanisms involved can be highly dynamic and context-dependent. It is important to use sensitive and high-throughput methods that can capture the complexity of these interactions and provide a comprehensive view of the molecular changes that occur during infection.

RNA sequencing enables high throughput analysis of the transcriptome landscape of cells. In host cells infected with pathogens, two organisms interact with markedly distinct transcriptomes. Dual RNA sequencing is a powerful in silico analysis method that enables the simultaneous study of the gene expression responses of both pathogens and host cells from the same samples, thereby deepening our understanding of their interaction [22,23]. In this study, we aimed to understand how SCN invasion modulates soybean gene expression, while simultaneously examining pathogen reactions across multiple hosts (e.g., compatible, semi-compatible, semi-incompatible, incompatible soybean). By utilizing dual RNA sequencing, we provide novel insights into the intricate interplay between host and parasite, revealing a diversity of defense mechanisms in soybean and virulence genes in the soybean cyst nematode. This study marks the first time that such a method has been employed to investigate soybean–SCN interactions, offering new insights into this crucial area of research.

## 2. Results

### 2.1. Greenhouse SCN Bioassay and Genotyping Rhg1 and Rhg4 Copy Number Variation (CNV)

The resistance of the lines PI 88788, PI 437654, PI 548402, and Lee 74 were characterized to their level of resistance according to the scale proposed by Schmitt and Shannon 1992 [24] (Table 1). The average number of females on Lee 74 was 110.2, and the highest to the lowest female index (FI) value was 63.0, 10.0, and 0.0 on PI 88788, PI 548402, and PI 437654, respectively (Figure 1). However, despite high levels of SCN replication on PI 88788 following inoculation with SCN HG type 1.2.5.7, the egg production was significantly lower than that of Lee 74 (Figure 1). Considering these results, in this study we called PI 437654 an incompatible line, PI 548402 a semi-incompatible line, PI 88788 a semi-compatible line, and Lee 74 a compatible line.

Based on the results mentioned above, which include the FI values and the number of eggs in the tested PI lines, we decided to focus primarily on the PI 437654 defense mechanism. This line has shown complete resistance against SCN HG type 1.2.5.7. To validate the SCN-responsive genes of PI 437654, we utilized PI 548402 and PI 88788, which have strong and weak defense mechanisms, respectively, against SCN HG type 1.2.5.7.

We evaluated the copy number variation (CNV) of the *rhg1* and *Rhg4* genes, well-studied loci conferring resistance to SCN, in all of the tested PI lines. To do this, we developed a Taqman assay from the conserved regions of these genes. The assay results indicated that the CNV of the *rhg1* locus was 8, 3, 3, and 1 in PI 88788, PI 548402, PI 437654, and Lee 74, respectively (Figure 2a). Additionally, the CNV of *Rhg4* was 1, 2, 3, and 1 in these same lines, respectively (Figure 2b).

### 2.2. Dual Transcriptome Sequencing and Assembly in Host and Nematode

We conducted a comprehensive analysis of the transcriptomic response of soybean to SCN, along with the reaction of SCN to different soybean hosts. RNA sequencing was performed on the four soybean PI lines, the three resistant lines (PI 88788, PI 548402, and PI 437654), and the susceptible line (Lee 74) at 5 and 10 days post-inoculation (dpi) with SCN HG type 1.2.5.7. Prior to conducting RNA sequencing, we confirmed infection of the roots with SCN HG type 1.2.5.7 at 5 and 10 dpi by amplifying the SCN 18S ribosomal gene from DNA extracted from the roots (Appendix A). Global patterns of gene expression in both soybean and *H. glycines* were evaluated using high-throughput RNA sequencing, with each sample having a high coverage of at least 50 million reads. In total, we obtained 58 million reads per sample from both infested and control conditions, of which approximately 87.83% were successfully mapped to the soybean reference transcriptome (Appendix A). When examining SCN HG type 1.2.5.7 at 5 dpi, we found that Lee 74 had the highest total number of reads mapped to *H. glycines*, while PI 437654 had the lowest (Figure 3, Appendix A). At 10 dpi, the total number of reads mapped to *H. glycines* had significantly decreased in Lee 74, PI 548402, and PI 88788. The highest number of reads was obtained from PI 88788 (Figure 3).

### 2.3. Intra- and Inter-Genotype Analyses Identify Pathways Pertaining to Defense in Each Resistant Line

In this study, we investigated within and between genotypes to first identify SCN-responsive genes and then confirm identified differentially expressed genes in PI 437654 (as a robust resistance line (FI = 0%)) using PI 548402 and PI 88788. We used resistant lines with different resistance levels to find promising SCN-responsive genes.

When examining differentially expressed genes (DEGs) within each genotype under infested and non-infested conditions, we identified that 167, 1966, 581, and 161 genes (DEGs) were upregulated (with cut-offs of log_2_ fold change ≥ 1, fragments per kilobase (kb) of transcript per million mapped reads (FPKM) ≥ 1, and *p* < 0.05) in PI 437654, PI 548402, PI 88788 and Lee 74, respectively, at 5 dpi (Figure 4). At 10 dpi, the number of DEGs significantly decreased in PI 437654, PI 548402, and PI 88788 but the number of DEGs remarkably increase in Lee 74 (Figure 4).

Intra-genotype gene ontology enrichment (GOE) analyses revealed functional categories, defined by high-level GO terms, that are involved in the immunity response of each line. In PI 437654, 44 out of 166 upregulated genes were found to be involved in the over-represented GO terms at 5 dpi (Table 2). Based on GO biological process analyses, these upregulated genes in PI 437654 at 5 dpi clustered into four groups: (1) programmed cell death in response to reactive oxygen species (ROS), (2) cell detoxification or response to stress, (3) carbon fixation, and (4) salicylic acid biosynthesis (Appendix A). At 10 dpi, 65 genes out of 145 upregulated genes were assigned to over-represented GO terms (Table 3). These genes clustered in three groups: (1) detoxification or response to stress, (2) salicylic acid biosynthesis, and (4) sulfur and glycosyl metabolic process (Appendix A).

Inter-genotype analyses of DEGs that were significantly upregulated in all lines at 5 dpi revealed that PI 437654, a strong source of resistance (FI = 0%), has 63 unique genes that were not differentially expressed in other resistant lines, PI 548402, PI 88788, with lower levels of resistance, FI = 10.0% and 63.0, respectively, or in Lee 74, the susceptible line (Figure 5). In addition, 42 SCN-responsive genes were detected in PI 437654, which were also found to be upregulated in PI 548402 (Figure 5). Comparison of PI 437654 with PI 548402 and PI 88788 resulted in the identification of 44 differentially upregulated genes that overlapped among these three resistant lines (Figure 5). Although our focus was on PI 437654, through this comparison we validated those SCN-responsive genes that were also observed in PI 548402 or/and PI 88788.

To gain insight into the functional categories of genes that are differentially expressed in response to SCN infection, we conducted GO enrichment analyses on uniquely differentially expressed genes in PI 437654, as well as on those genes that overlapped with PI 548402 and PI 88788 (Table 4 and Table 5). We identified several over-represented GO term categories and corresponding genes involved in these categories at 5 dpi. In PI 437654, six GO term categories were identified among the 63 uniquely upregulated genes (Figure 5), including response to chemical, negative regulation of development and reproductive process, photosynthesis and carbon fixation, salicylic acid metabolic process, oxazole or thiazole metabolic process, and fructose1,6 bisphosphate metabolic process (Appendix A). At the same time point, 16 genes out of 42 significantly upregulated genes detected in both PI 437654 and PI 548402 were found to be enriched in four distinct GO terms, including response to chemical and oxidative stress, programmed cell death in response to ROS, cellular response to an inorganic substance, and response to ethylene (Table 4 and Appendix A and Figure 5). In addition, we identified 20 SCN-responsive genes overlapped in PI 437654, PI 548402 and PI 88788 and categorized them into two over-represented GO terms: response to stress and cellular detoxification, and regulation of primary metabolic processes such as RNA and DNA (Table 4 and Appendix A and Figure 5). Finally, we found that among the six common DEGs detected in both PI 437654 and PI 88788 at 5 dpi, only one gene (*Glyma_02G028000*) was categorized into an over-represented GO, specifically the collagen catabolic process (Table 4 and Figure 5). These results provide valuable insights into the functional categories of genes that are involved in the soybean response to SCN infection and can help inform future studies aimed at improving soybean resistance to SCN.

At 10 dpi, inter-genotype analyses revealed PI 437654 as a highly resistant source (FI = 0%), displaying 39 unique genes not differentially expressed in other resistant lines, including PI 548402 and PI 88788, which showed lower resistance levels with FIs of 10.0% and 63.0%, respectively, and the susceptible line, Lee 74 (Figure 5). Additionally, four SCN-responsive genes were identified in PI 437654 that were also upregulated in PI 548402, while 31 SCN-responsive genes were upregulated in PI 88788 (Figure 5). By comparing PI 437654 with PI 548402 and PI 88788, we identified 24 differentially upregulated genes that were shared among all three lines (Figure 5). This comparison allowed for validation of a portion of the SCN-responsive genes identified in PI 437654, using PI 548402 and PI 88788 as additional reference points.

At 10 dpi, GO enrichment analysis identified four enriched genes out of thirty-nine unique DEGs in PI 437654, which were annotated in the GO terms ammonium transmembrane transport and sulfur amino acid biosynthetic process (Table 5 and Appendix A). Furthermore, among the four common genes shared between PI 437654 and PI 548402, two were identified as part of a set of genes enriched in GO terms related to oxazole or thiazole biosynthetic process, S-glycoside catabolic process, and sulfur compound metabolic process (Appendix A). GO enrichment analyses identified 14 genes out of 31 SCN-responsive genes that overlapped in PI 437654 and PI 88788 (Table 5 and Figure 5), which were categorized into five over-represented GO terms: salicylic acid biosynthetic process, monocarboxylic acid biosynthetic process, cellular oxidant detoxification, organic acid metabolic process, regulation of nucleic acid-templated transcription, and response to stress (Appendix A). Among PI 437654, PI 548402, and PI 88788, there were 24 overlapped genes that we were not able to assign to any enriched GO terms (Table 5).

The pathway analysis of DEGs in PI 437654, PI 548402, PI 88788, and Lee 74 revealed enriched immune response pathways in resistant lines. KEGG analysis indicated that the MAPK signaling pathway, phenylpropanoid biosynthesis, plant hormone signal transduction, and biosynthesis of secondary metabolite were significant pathways associated with soybean immune response to SCN HG Type 1.2.5.7 in PI 437654, PI 548402, and PI 88788 at 5 dpi (Appendix A). In PI 437654, we observed that there are other pathways associated with the immune systems, such as plant–pathogen interaction, carbon fixation in the photosynthetic organism, and carbon metabolism. These pathways set PI 437654 apart from PI 548402 and PI 88788, making it a unique SCN source (Appendix A). Pathway analysis of PI 437654 at 10 dpi revealed that upregulated DEGs were enriched in several metabolic pathways, including phenylpropanoid biosynthesis, isoflavonoid biosynthesis, flavonoid biosynthesis, thiamine metabolism, glutathione metabolism, sulfur metabolism, cysteine and methionine metabolism, cyanoamino acid metabolism, circadian rhythm, biosynthesis of cofactors, biosynthesis of secondary metabolites, ubiquinone and terpenoid–quinone biosynthesis (Appendix A).

### 2.4. Interactions between SCN Transcriptomes and Soybean Proteins

In order to study the expression of SCN pathogenicity genes during the infection process and assess the behaviour of SCN towards different hosts, we analyzed the transcriptomes of SCN HG type 1.2.5.7 at two time points in the four soybean PI lines. Comparison of the SCN transcriptomes across different soybean lines showed that the pathogenicity gene expression can be altered by the host plant. Furthermore, a two-by-two comparison of the host plants revealed that the SCN HG type 1.2.5.7 transcriptome in interaction with Lee 74 exhibited the largest number of differentially expressed genes compared with the transcriptome in PI 437654. In contrast, the transcriptome of SCN HG type 1.2.5.7 in interaction with Lee 74 had the highest similarity in gene expression profile with PI 88788 (Figure 6). For instance, the SCN transcriptome in Lee 74 had 123 differentially upregulated and 136 differentially downregulated genes when compared with its transcriptome in PI 437654, while only one gene was differentially upregulated in Lee 74 when compared with PI 88788 (Figure 6). This suggests that SCN adapts its gene expression profile in response to the host plant and that different soybean lines may have different levels of resistance against SCN.

Effector genes play a crucial role in the molecular interactions between SCN and soybean. These genes encode secretory proteins that are translocated into the root tissue of soybean during infection. In this study, our primary focus was on identifying and characterizing these effectors. To accomplish this, putative effectors were extracted from the differentially expressed genes in four soybean lines, PI 437654, PI 548402, PI 88788, and Lee 74. To identify putative effectors in SCN, we considered the presence of secretory signal peptide and the likelihood of extracellular localization. Analysis of putative effectors revealed that PI 437654 had a unique set of effectors, unlike the other soybean hosts. Furthermore, the putative effectors of PI 548402 shared a 62% similarity with those of PI 88788 and 50% similarity with Lee 74. The most commonly shared effector genes were found between PI 88788 and Lee 74, with 89% similarity (Table 6). These findings highlight the importance of effectors in the pathogenesis of *H. glycines* and provide insight into the diversity and conservation of these genes in different soybean hosts.

## 3. Discussion

The main objective of this study was to investigate the defense mechanism of PI 437654, a highly resistant soybean line (FI = 0%), against SCN HG type 1.2.5.7. PI 548402, and PI 88788, which exhibit lower levels of resistance (FI = 10% and 63%, respectively), were also included to validate resistant genes and identify the resistance genes unique to PI 437654. Furthermore, the roots of these three resistant lines, as well as the susceptible line Lee 74 (FI = 100%), were analyzed to study changes in soybean root transcriptome in response to SCN. To identify candidate genes responsible for resistance in PI 437654, GO enrichment analyses were employed, which allowed for the extraction of gene sets involved in the enriched biological process. These findings provide insight into the mechanisms underlying resistance to SCN, particularly in PI 437654, and can aid in the development of more resistant soybean cultivars. Simultaneously, dual RNA sequencing allows us to examine SCN reactions across multiple soybean hosts (e.g., compatible, semi-compatible, semi-incompatible, incompatible) to study the expression of SCN pathogenicity genes during the infection process.

In this research, a long-term SCN stress at 5 and 10 dpi was chosen for RNA sequencing analysis to investigate the dual transcriptome reaction between soybean and SCN. Hammond et al. (2004) [27] classified genes that respond to stress into two categories: “early” and “late” genes. The “early” genes exhibit a rapid response and are generally non-specific to target stresses, while the “late” genes have a delayed expression but can have a significant impact on the morphology, physiology, and/or metabolism of plants. Furthermore, these “late” genes are often specific to target stresses. According to Matsye et al. (2011) [28], Peking-type resistance is characterized by a rapid and potent resistant reaction that results in the formation of a necrotic region around the syncytium by 5 dpi. On the other hand, PI 88788-type resistance is a prolonged but potent resistant reaction, which does not show any cytological evidence of a reaction at 5 dpi. Previous studies on cell fate against SCN have found that selecting “late” genes increases the likelihood of identifying specific genes. Consequently, 5 and 10 dpi were used for dual transcriptome reaction between soybean and SCN. The results of our dual transcriptome analysis provide valuable insight into the complex reaction of soybean gene expression in response to SCN parasitism. Furthermore, the coordinated expression of SCN HG type 1.2.5.7 genes potentially involved in parasitism against different soybean lines can now be better understood.

According to the total number of reads mapped to *H. glycines*, results indicated that SCN HG type 1.2.5.7 had the lowest penetration in PI 437654, possibly due to a strong passive defense mechanism, while it had the highest penetration in Lee 74, likely due to a physical barrier weakness. A significant decrease in the frequency of reads in Lee 74 at 10 dpi suggested that SCN HG type 1.2.5.7 had overcome the defense mechanism of Lee 74, leading to successful development into adult females and males. In PI 548402, the observed reduction in read frequency may be attributed to the strong two-layered immunity system, which overcomes SCN HG type 1.2.5.7. It is possible that the second-stage juveniles (J2) nematode cannot form specialized feeding sites, syncytia, and, therefore, are unable to copulate, leading to their eventual starvation and death. At 10 dpi, while the total number of reads mapped to *H. glycines* in PI 88788 was significantly reduced, it still remained relatively large. This observation may suggest that there was a battle between the SCN and PI 88788, where neither was able to fully overcome the other.

Comparing differentially expressed genes (DEGs) within each genotype under infested and non-infested conditions at two time points, 5 and 10 dpi, demonstrated that the number of DEGs significantly decreased in PI 437654, PI 548402, and PI 88788 at 10 dpi. In contrast, the number of DEGs remarkably increase in Lee 74, which may be due to the ability of SCN effector proteins to hijack and alter the gene expression patterns of Lee 74, leading to the induction of a large number of DEGs.

Host transcriptional pattern reprogramming is often triggered by pathogen invasion. Plasma membrane Ca channels are among the early sensors that respond to pathogen attacks by increasing Ca influx into the cell cytoplasm, as reported by Sun et al. (2015) [29]. Ca2+ sensor proteins, such as calmodulin, EF-hand domain, and Ca2+-dependent protein kinases (CDPKs), detect the transient increase in Ca^2+^ signatures, as observed by Houqing Zeng et al. (2017) [30] and Sun et al. (2015) [29]. Calmodulin, despite lacking enzymatic activity, binds to calmodulin-binding proteins, thereby stimulating the synthesis and accumulation of SA during immunity, as highlighted by Choudhury et al. (2017) [31] and Gilroy et al. (2016) [32]. SA, in turn, activates systemic acquired resistance (SAR), providing broad-spectrum and long-lasting resistance against pathogens [29]. The study identified calmodulin-binding proteins *(Glyma.19G229500*, *Glyma.03G232400*, and *Glyma.19G229400)* at 5 dpi and calmodulin-binding proteins (*Glyma.19G229500*, *Glyma.05G237200*, *Glyma.07G093900*, and *Glyma.09G182400*) at 10 dpi in PI 437654, but not in Lee 74. *Glyma.03G232400* and *Glyma.19G229400* were unique to PI 437654, and *Glyma.05G237200* and *Glyma.07G093900* were also observed in PI 88788 (Figure 7). *Glyma.19G229500* and *Glyma.09G182400* were validated in both PI 548402 and PI 88788, with *Glyma.19G229500* being differentially expressed at both time points and validated by other resistant lines. These findings align with previous studies by Kofsky et al. (2021) [33] and Zhang et al. (2017) [34] that reported the presence of calmodulin-binding proteins in transcriptome comparisons of different genotypes under SCN HG type 2.5.7-treated conditions. Additionally, the study detected two differentially expressed EF-hand domain proteins (*Glyma.19G160100* and *Glyma.16G214800*) in PI 437654 at 5 dpi, but not in other resistant lines (Table 2 and Figure 7).

Protein kinase is a critical component of intracellular signal transduction and plays an essential role in stress response [35]. In current research, three proteins with protein kinase activity, namely, *Glyma.07G184000*, *Glyma.17G173000*, and *Glyma.18G219600*, were identified at 5 dpi. Among them, *Glyma.07G184000* and *Glyma.17G173000* were validated using both PI 548402 and PI 88788, while *Glyma.18G219600* was only observed as differentially expressed in PI 437654. These protein kinases identified as SCN-responsive genes in the current study are aligned with the observations made by Ithal et al., 2007 [36], and Han et al., 2015 [37], which also detected these genes in response to SCN HG type 0 and HG type 1.2.3.5.7 [36,37].

Transcription factors play crucial roles in signal transduction by activating or suppressing the expression of defense genes and regulating the crosstalk between different signaling pathways. As they bind to specific cis-acting elements in gene promoters, transcription factors are positioned at the penultimate step of the signal cascade to directly control the downstream target gene expression. In the current study, several proteins with AP2/ERF domains, such as *Glyma.03G162400*, *Glyma.03G162700*, *Glyma.05G186700*, *Glyma.10G186800*, *Glyma.13G122500*, *Glyma.13G123100*, *Glyma.19G163700*, and *Glyma.19G163900*, which have transcription regulator activity, were found to be differentially upregulated in PI 437654 at 5 dpi. These genes were confirmed to play an important role against SCN HG type 1.2.5.7 through their expression in other resistant lines, including PI 548402 and PI 88788. The presence of AP2 is aligned with the observations made by Kofsky et al. [33], who studied resistant and susceptible resources from *G. soja* against SCN HG type 2.5.7. Another upregulated transcription factor in all three resistant lines at 5 dpi is the WRKY protein (e.g., *Glyma.08G018300*). At 10 dpi, six WRKY genes, including *Glyma.04G223300*, *Glyma.13G267600*, *Glyma.17G222300*, *Glyma.13G267500*, *Glyma.14G103100*, and *Glyma.18G213200*, were detected in PI 437654. Of these, the first three were also observed in PI 88788, and the last three were validated in both PI 548402 and PI 88788. The upregulation of WRKY genes as SCN-responsive genes in this study is supported by Zhang et al., 2017 [34], as well as the observations made by Ithal et al., 2007 [36], Han et al., 2015 [37], and Kofsky et al., 2021 [33].

Plants encounter various stresses and ROS such as hydrogen peroxide (H_2_O_2_), superoxide anions (O_2_^•−^), hydroxyl radical (^•^OH), and singlet oxygen (^1^O_2_) play important roles in signal transduction. However, excessive ROS accumulation can be harmful and even lead to cell death. Thus, a delicate balance between ROS production and ROS-scavenging pathways must be maintained [38]. In the present study, at 5 dpi, several peroxidase genes including *Glyma.19G011800*, *Glyma.01G130500*, *Glyma.03G038300*, *Glyma.03G038500*, *Glyma.09G023000*, *Glyma.09G057100*, *Glyma.15G052700*, and *Glyma.20G001400* were found to be differentially expressed in PI 437654, but not in Lee 74. Among these peroxidase genes, *Glyma.01G130500*, *Glyma.03G038500*, *Glyma.09G023000*, *Glyma.09G057100*, and *Glyma.20G001400* were observed in both PI 548402 and PI 88788, while *Glyma.19G011800* and *Glyma.03G038300* were only validated in PI 548402. *Glyma.15G052700* was the only unique peroxidase gene that was differentially expressed in PI 437654 at 5 dpi. At 10 dpi, *Glyma.09G277900*, *Glyma.20G169200*, and *Glyma.11G161600* were identified as peroxidase genes, while *Glyma.05G161300* was differentially expressed and annotated as a glutathione S-transferase (GST). The first two peroxidase genes were confirmed using PI 88788, while the last peroxidase and GST were only identified in PI 437654. These findings are in agreement with the observation by Miraeiz et al., 2020 [39], who studied RNA-Seq profiling of Peking, Fayette, Williams 82, and a wild relative (*Glycine soja*, PI 468916) against SCN HG type 0.

Oxidative enzymes are pivotal to plant metabolism, catalyzing a wide array of reactions involved in hydroxylation, DNA repair, and post-translational modification, as well as the activation and catabolism of plant growth regulators. The 2OG-Fe(II) oxygenase superfamily and cytochrome P450 (CYP) are two important classes of these enzymes [40]. In this investigation, four genes belonging to the 2OG-Fe(II) oxygenase superfamily (*Glyma.03G096500*, *Glyma.07G124400*, *Glyma.08G169100*, and *Glyma.14G058700*) were identified at 5 dpi, and two genes (*Glyma.04G227900* and *Glyma.18G273200*) were detected at 10 dpi in PI 437654, but not in Lee 74 (Figure 7). Additionally, 11 genes annotated as Cytochromes P450 were identified at 10 dpi in PI 437654 (e.g., *Glyma.01G135200*, *Glyma.02G156100*, *Glyma.03G143700*, *Glyma.05G022100*, *Glyma.09G049200*, *Glyma.10G114600*, *Glyma.11G062500*, *Glyma.11G062600*, *Glyma.11G062700*, *Glyma.15G156100*, and *Glyma.16G195600*) (Table 3). This study’s results reveal that only two P450 genes (*Glyma.04G035600* and *Glyma.09G144300*) were validated and differentially expressed in PI 548402 but not in Lee 74 (Figure 7 and Table 4), implying that these genes play a crucial role in soybean defense against SCN infection. Our findings are in agreement with previous studies by Ithal et al., 2007 [36], and Han et al., 2015 [37], further highlighting the importance of oxidative enzymes in the plant’s defense response.

The plant cell wall can serve as a protective and physical barrier for limiting pathogen penetration into the plant cell [41,42]. Lignin, a three-dimensional polymer, is a major component of plant cell walls, composed of monomeric units such as syringyl, guaiacyl, and p-hydroxyphenyl, derived from phenylalanine in the cytoplasm and transported to the cell wall. The monomeric units are polymerized by laccase and peroxidase enzymes, which catalyze the random cross-linking necessary for the formation of the lignin polymer [43,44,45]. Polymerized lignin reinforces the strength and rigidity of plant cell walls and is a key component of the plant’s response to environmental factors [46,47]. In this study, nine differentially expressed peroxidase genes were over-represented in PI 437654 at 5 dpi (Table 2). Based on KEGG analyses, these genes are involved in the phenylpropanoid pathway and are directly responsible for producing syringyl lignin, guaiacyl lignin, 5-hydroxy-guaiacyl lignin, and p-hydroxyphenyl lignin. Among these genes, *Glyma.01G130500*, *Glyma.03G038500*, *Glyma.09G023000*, *Glyma.09G057100*, and *Glyma.20G001400* were validated through PI 548402 and PI 88788 and were not found to be differentially expressed in Lee 74. *Glyma.03G038300* and *Glyma.19G011800* were only validated in PI 548402 and were not observed in the susceptible line (Figure 7 and Table 4). Another peroxidase gene that was differentially expressed at 5 dpi was *Glyma.15G052700*, which was only observed in PI 437654 and not in the susceptible line (Figure 7 and Table 4). At 10 dpi, one differentially expressed laccase gene and nine peroxidase genes were detected. *Glyma.01G108200* was annotated as a laccase gene and was validated by both PI 548402 and PI 88788. Out of the nine peroxidase genes, only *Glyma.09G277900* and *Glyma.20G169200* were not found to be differentially expressed in Lee 74 and were validated only by PI 88788 (Figure 7 and Table 5). Another peroxidase gene that was differentially expressed at 10 dpi was *Glyma.11G161600*, which was only observed in PI 437654 and did not significantly upregulate in Lee 74 (Figure 7 and Table 3). In addition, dirigent is another cell-wall-related gene that modulates cell wall metabolism [48]. At 10 dpi, Glyma.19G151200, which was annotated as a dirigent gene, was not detected in the susceptible line and was validated in PI 88788 (Figure 7). Our finding about the role of peroxidase, laccase, and dirigent in cell wall rigidity against SCN invasion is supported by Afzal et al., 2009 [49], which compared two NILs including rhg1rhg1/Rhg4Rhg4 and Rhg1Rhg1/Rhg4Rhg4 against SCN HG type 0 and Miraeiz et al., 2020 [39] who studied different soybean lines against SCN HG type 0. Glycoside hydrolase is another gene involved in cell wall polysaccharide metabolism [50]. Among the DEGs observed during SCN invasion in PI 437654, *Glyma.12G053900* at 5 dpi and *Glyma.11G129300*, *Glyma.12G054200*, and *Glyma.13G346700* at 10 dpi were upregulated. All of these genes are annotated as glycoside hydrolases. *Glyma.12G053900* was observed in both PI 548402 and PI 88788, and *Glyma.11G129300* was validated through PI 548402. Lipoxygenase is another gene that induces cell wall modification to limit pathogen invasion [51], and at 10 dpi, the lipoxygenase gene *Glyma.19G263300* was detected in PI 437654 and validated in PI 88788.

Interestingly, the rhg1 genes (e.g., *Glyma18g02580*, *Glyma18g02590*, and *Glyma18g02610*) and Rhg4 gene (*Glyma.08G108900*), which are promising genes known to confer SCN resistance to HG type 0, were not identified as significantly different in the current study, consistent with the findings of Zhang et al., 2017 [34]. Additionally, the pathways of plant–pathogen interaction, carbon fixation in the photosynthetic organism, and carbon metabolism, which were the top three enriched pathways for upregulated DEGs in PI 437654, were not observed in other resistant lines. This finding is consistent with the observation by Shi et al., 2021 [52], who studied PI 437654 against SCN HG type 1.2.3.5.7. However, this result is in contrast to the reports of Zhang et al., 2017 [34], who studied *Glycine soja* interaction with SCN HG type 2.5.7 and identified genes involved in carbon fixation and photosynthesis pathways as remarkably downregulated. According to Shi et al., 2021 [52], a large number of DEGs that were upregulated in the incompatible soybean variety PI 437654 were involved in the plant hormone pathway, MAPK signaling, and phenylpropanoid biosynthesis pathway, which is in agreement with the results of the current study on PI 437654.

Dual RNA sequencing has provided a significant advantage in investigating the interaction between pathogen and host simultaneously. Through transcriptome analysis of SCN HG type 1.2.5.7, novel secreted effectors with potential roles in the SCN–host interaction have been identified. The establishment and maintenance of the syncytium, a crucial step for the long-term parasitic success of SCN, involves multiple stages such as hatching stimuli, host attraction, root penetration, tissue modification, feeding site formation, and immune system suppression. The pathogenicity and severity of SCN are dependent on the successful completion of these stages, as well as the pathogen’s classification as pathogenic or non-pathogenic. This study represents the first investigation of the SCN transcriptome in different hosts with varying levels of resistance, shedding light on the virulence strategies employed by SCN to overcome resistant hosts. Notably, 51 putative effectors showing differential expression patterns in both resistant and susceptible lines were identified, with 39 of them being newly discovered. Furthermore, only a limited overlap was observed between the effectors identified in PI 437654 and those identified in other resistant and susceptible lines, highlighting the adaptive ability of SCN to modulate its virulence genes to overcome resistant hosts.

## 4. Materials and Methods

### 4.1. Soybean and SCN Procurement for Inoculation

The SCN HG type 1.2.5.7 inbred strain provided by the University of Illinois was maintained for more than 50 generations. To extract eggs, cysts were first sterilized by briefly treated with 0.5% sodium hypochlorite for approximately 5 s, followed by thorough rinsing with tap water. Eggs were extracted by crushing cysts with a rubber stopper and collecting them using a nested sieve consisting of 75 µm (mesh 200) and 25 µm (mesh 500) sieves (USA Standard Testing Sieve), following which the number of eggs per unit volume was determined by stirring the egg suspension with a magnetic stir bar and counting an aliquot of the suspension using a 10× microscope on a chambered counting slide. This process was repeated eight times to estimate the number of eggs per unit volume.

Seeds of PI 88788, PI 437654, PI 548402, and Lee-74 were procured from Agriculture and Agri-Food Canada, Harrow, Canada. The soybean seeds were surface-sterilized by soaking in 5% (*v/v*) sodium hypochlorite for 5 min and washed with sterilized water three times for 2 min each time. The sterilized seeds were then germinated between moist sterilized filter papers in a petri dish in dark conditions and at 25 °C. After three days, the seedlings were transferred to prepared cons that contained a cotton ball, turface, and a mix of two types of sand (a 50:50 mix of masonry sand and beach sand) from bottom to top. One day following the transfer, each plant underwent inoculation with 4000 eggs of SCN HG type 1.2.5.7. As for the control plants, an equivalent volume of water was added instead. Subsequently, all the inoculated and control plants were placed in a growth chamber located at the University of Guelph for incubation. The chamber was set to a day/night temperature of 27 °C/25 °C with a photoperiod of 16 h and light intensity of 450 umoles/m2/s, and the relative humidity was set at 60%. The seedlings were watered daily to maintain plant moisture. This experiment followed a complete block design (CRD) with three biological replications.

### 4.2. Female Index (FI) Calculation

At 52 dpi, the cysts were thoroughly washed with water through an 850 µm (mesh 20) sieve stacked on a 250 µm (mesh 60) sieve (USA Standard Testing Sieve, Lewis Center, OH, USA). The female index (FI) was calculated on a per-plant basis using the following equation:Female Index (FI) = (Number of cysts on test line)/(Number of cysts on Lee 74) × 100

Following the cyst count and FI% calculation, eggs were released from the cysts and counted under a 10× microscope (Olympus Canada Inc., Richmond Hill, ON, Canada).

### 4.3. Rhg1 and Rhg4 Copy Number Variation

To determine the copy number of *rhg1* and *Rhg4*, the GmSNAP18 and GmSHMT08 genes were used as references, following the protocols described by Kadam et al., 2016 [53] and Patil et al., 2019 [54]. Oligonucleotide probes specific for *rhg1* and *Rhg4* were labelled with FAM at the 5′ ends, while the probe for the reference gene, lectin (*Le1*), was labelled with the fluorescent dye VIC at the 5′ ends. The 3′ ends of all probes were labelled with the quencher dye MGBNFQ (Appendix A; Applied Biosystems, Foster City, CA, USA).

Real-time PCR was performed using a 10 μL reaction mixture containing 5 μL 2x TaqMan Universal PCR Master Mix (Applied Biosystems), 500 nM of each primer, 100 nM of each probe, and 15ng of genomic DNA. Two technical replicates were conducted for each sample, and template-free or negative controls were included. The QuantStudio 6 flex System (Applied Biosystems, CA, USA) was used to run the real-time PCR program, consisting of an initial denaturation step of 10 min at 95 °C, 40 cycles of 15 s at 95 °C, and 1 min at 60 °C. The copy number assay was calibrated using Williams 82 as a control sample, which carries a single copy of insertion. Copy number analysis of the target genes was performed using the CopyCaller Software v2.0 (Applied Biosystems), following the manufacturer’s instructions.

### 4.4. Tissue Collecting and Infection Confirmation

The current study aimed to identify specific SCN-responsive genes in soybean and host-specific pathogenesis genes in *H. glycines*, and 5 and 10 dpi were chosen as the optimal time points based on a previous study [27,28]. Root samples were collected from both inoculated and control seedlings, rapidly washed, and flash-frozen quickly using liquid nitrogen. The samples were then stored at −80 °C for subsequent analysis. To confirm the infection of the roots with SCN HG type 1.2.5.7 at 5 and 10 dpi, PCR amplification of the nematode 18S ribosomal gene was performed using root-extracted DNA (Appendix A).

### 4.5. RNA Extraction and Dual RNA Sequencing

Total RNA was extracted from both soybean and SCN using the Invitrogen™ PureLink™ RNA Mini Kit, following the manufacturer’s instructions. The extracted RNA samples were evaluated for both quantity and quality at the Genomics Facility of the University of Guelph (Guelph, ON, Canada) using an Agilent 2100 BioAnalyser System (Agilent Technologies, Palo Alto, CA, California). Only high-quality RNA samples (RNA Integrity Number ≥ 9) were selected for sequencing at the Genome Quebec, Innovation Center of McGill University (Montreal, QC, Canada).

The raw data were subjected to quality control checks, which involved removing reads containing adaptor sequences, poly-N sequences, and low-quality reads. Reference genome and gene model annotation files were obtained from the Phytozome website [55] for soybean and WormBase Parasite website [56,57] for SCN.

The reads were aligned to the reference genome using Hisat2 v. 2.0.5 [58], and FeatureCounts v. 1.5.0-p3 [59] was utilized to quantify the number of reads mapped to each gene. Gene expression levels were measured in FPKM values. Differentially expressed genes were identified using DEBRowser/DESeq2 with log2 FC (fold change) of > 1 and a *p*-value of < 0.05.

### 4.6. Data Analyses and Gene Pathway Analyses

The jvenn software [60] was employed to generate Venn diagrams. For gene annotation analysis of soybean and SCN, the Biomart in EnsemblPlants database (accessible at https://plants.ensembl.org/, accessed on 1 July 2023) and Wormbase Parasite were used, respectively. GOE analysis, hierarchical clustering tree, network analysis, and KEGG were performed using ShinyGO (ShinyGO v0.61) [61].

### 4.7. Putative Effectors

To identify putative effectors among the differentially expressed genes in SCN, we evaluated the presence of secretory signal peptide and the probability of extracellular localization for all differentially expressed proteins. First, we extracted the protein sequences of each differentially expressed gene from Wormbase Parasites [56,57]. Then, we used SignalP 6.0 (https://services.healthtech.dtu.dk/service.php?SignalP, accessed on 1 July 2023) to predict signal peptides and cleavage sites, with default Eukaryote setting, based on the protein language model [26]. To further filter the candidate effectors, we utilized the DeepLoc 2.0 server (https://services.healthtech.dtu.dk/service.php?DeepLoc-2.0, accessed on 1 July 2023) [62] to predict extracellular localization. Proteins showing at least a 50% probability of containing a signal peptide signature and extracellular localization prediction were selected and considered as putative effectors.

## 5. Conclusions

Our study has uncovered the mechanisms underlying the robust resistance of PI 437654 against SCN HG type 1.2.5.7, which involves reinforcement of cell walls and enhancement of physical barriers that impede pathogen penetration. Notably, PI 437654 exhibits a lower number of defense-related genes, suggesting its ability to confront *H. glycines* at the root entry point. Our analysis of egg production in different hosts against SCN HG type 1.2.5.7 has led to the classification of PI 437654 as an incompatible line, PI 548402 as a semi-incompatible line, PI 88788 as a semi-compatible line, and Lee 74 as a compatible line, providing a spectrum of resistant lines for investigating the soybean–SCN interaction. By focusing on PI 437654 and validating with PI 548402 and PI 88788, we have identified several promising candidate genes for SCN resistance, including 2OG-Fe(II) oxygenase superfamily, cytochrome P450 (CYP), AP2 domain, WRKY, protein kinase, peroxidase, leucine-rich repeat, calmodulin-binding protein, laccase, and dirigent, consistent with previous studies. As SCN resistance is a complex trait, it is likely that multiple genes are required to confer resistance against *H. glycines*. The diverse behaviour of SCN HG type 1.2.5.7 in different hosts, as observed through transcriptome monitoring, validates the hypothesis of SCN’s adaptive ability to overcome resistance in different hosts. It is plausible that secreted proteins or effectors of SCN undergo accelerated evolution in response to strong selective pressure from host immunity. These findings may explain the shift in the SCN population in soybean fields where the same source of resistance has been repeatedly used.

Our study has uncovered new molecular mechanisms that underlie soybean’s resistance to SCN, and has deepened our understanding of the complex interplay between SCN and soybean hosts. These discoveries have far-reaching implications for the development of effective strategies to manage SCN and underscore the importance of breeding for resistance by targeting multiple genes. Future research in this area holds promise for identifying additional mechanisms and genes that contribute to SCN resistance, ultimately improving soybean production and sustainability in regions affected by SCN.

## Figures and Tables

**Figure 1 ijms-24-16232-f001:**
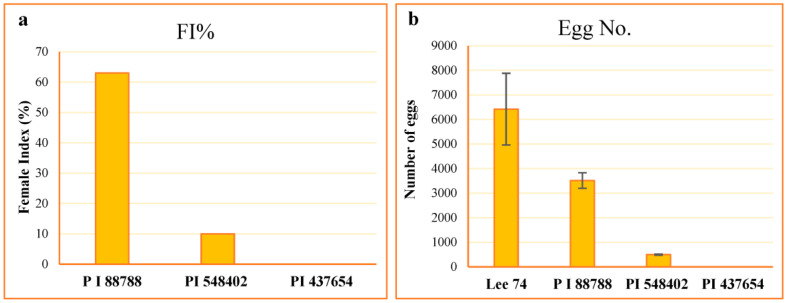
Estimation of FI values (**a**) and counting number of eggs ± SE (**b**) in each line.

**Figure 2 ijms-24-16232-f002:**
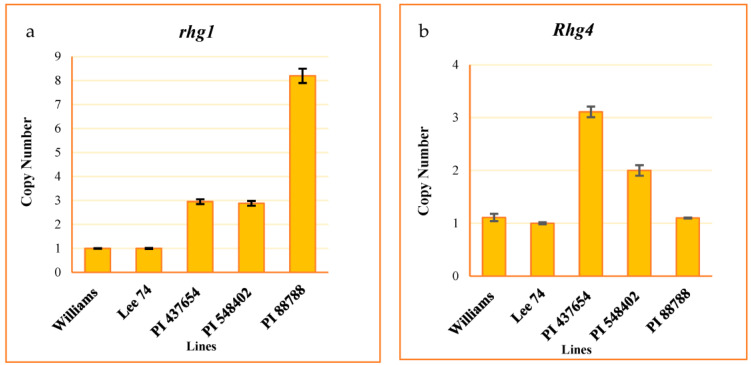
CNV of *rhg1* ± SE (**a**) and *Rhg4* ± SE (**b**). Williams used as a calibrator.

**Figure 3 ijms-24-16232-f003:**
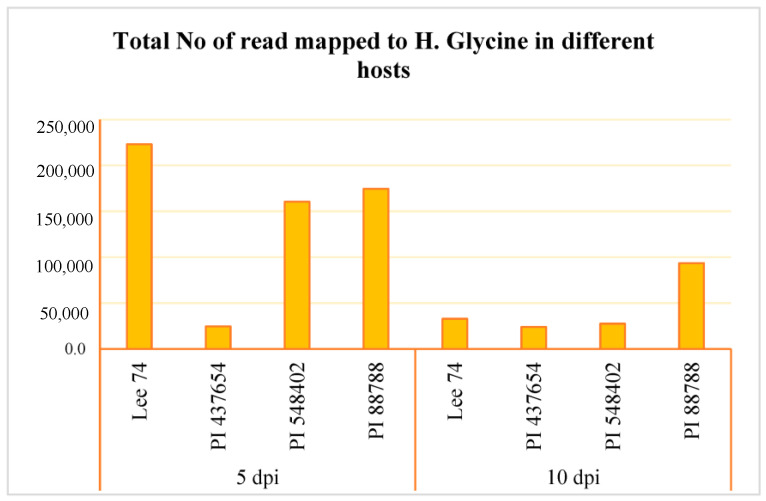
Total number of read mapped to *H. glycines* in four different hosts (e.g., Lee 74, PI 437654, PI 548402, and PI 88788) and at two time points (e.g., 5 and 10 dpi).

**Figure 4 ijms-24-16232-f004:**
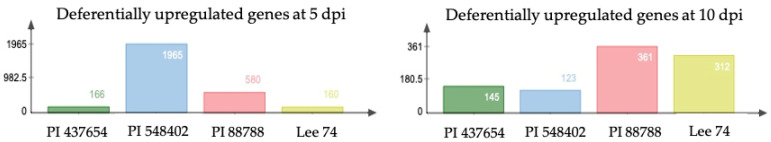
The reaction of differentially upregulated genes in PI 88788, PI 548402, PI 437654, and Lee 74 against SCN HG type 1.2.5.7.

**Figure 5 ijms-24-16232-f005:**
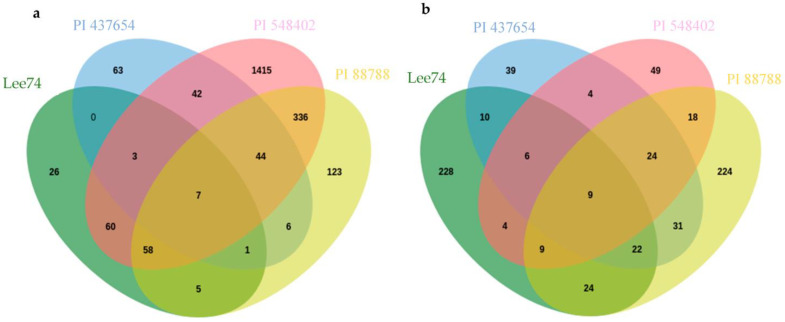
Venn diagram showing the number of unique and overlapping SCN-responsive sets of genes in PI 437654 compared to PI 548402, PI 88788, and Lee 74 which were upregulated at 5 dpi (**a**) and 10 dpi (**b**).

**Figure 6 ijms-24-16232-f006:**
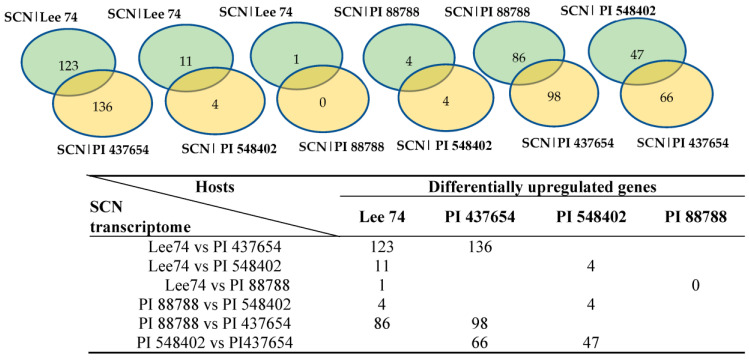
Number of genes differentially express in SCN HG type 1.2.5.7 transcriptome in reaction with the four different hosts.

**Figure 7 ijms-24-16232-f007:**
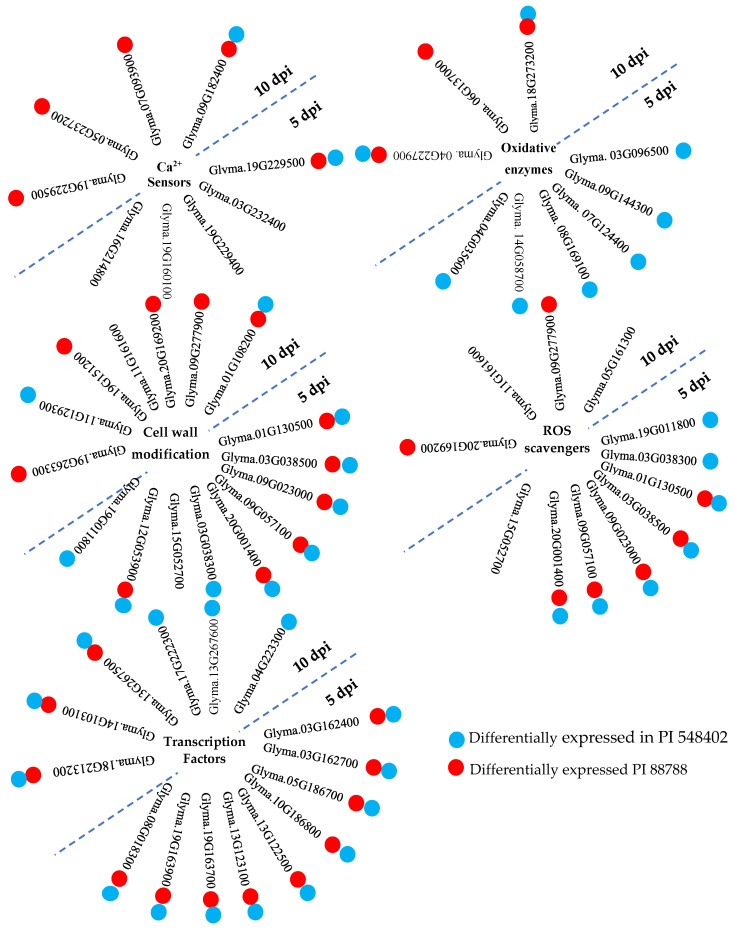
A summary of inter-genotype analyses of differentially expressed genes that were significantly upregulated in PI 437654 in response to SCN HG type 1.2.5.7 at 5 and 10 dpi and were not observed in Lee 74 as a susceptible line. Each cluster of genes is associated with specific functional categories involved in SCN-resistant defense mechanisms. Blue balls indicate SCN-responsive genes also expressed in PI 548402. Red balls indicate SCN-responsive genes also expressed in PI 88788.

**Table 1 ijms-24-16232-t001:** Female index scale of tested soybean lines against SCN HG type 1.2.5.7.

Female Index (%)	Rating	Label
<10	Resistant	R
≥10 to 30	Moderately resistant	MR
>30 to 60	Moderately susceptible	MS
>60	Susceptible	S

**Table 2 ijms-24-16232-t002:** PI 437654 upregulated genes that were assigned in over-represented GO terms in 5 dpi.

Gene Name	PFAM Description	InterPro Description
Glyma.03G242100	Copine	C2 domain
Glyma.19G239500	Copine	C2 domain
Glyma.03G232400	Calmodulin_bind	Calmodulin-binding protein 60
Glyma.19G229400	Calmodulin_bind	Calmodulin-binding protein 60
Glyma.19G229500	Calmodulin_bind	Calmodulin-binding protein 60
Glyma.04G035600	p450	cytochrome P450 (CYP)
Glyma.19G160100	EF-hand_7	EF-hand domain
Glyma.01G124500	Ferritin	Ferritin/DPS protein domain
Glyma.20G087000	Response_reg	GAF domain
Glyma.12G053900	Glyco_hydro_1	Glycoside hydrolase, family 1
Glyma.19G011800	peroxidase	Haem peroxidase
Glyma.01G130500	peroxidase	Haem peroxidase, Plant ascorbate peroxidase
Glyma.03G038300	peroxidase	Haem peroxidase, Plant ascorbate peroxidase
Glyma.03G038500	peroxidase	Haem peroxidase, Plant ascorbate peroxidase
Glyma.03G038600	peroxidase	Haem peroxidase, Plant ascorbate peroxidase
Glyma.09G023000	peroxidase	Haem peroxidase, Plant ascorbate peroxidase
Glyma.09G057100	peroxidase	Haem peroxidase, plant/fungal/bacterial
Glyma.15G052700	peroxidase	Haem peroxidase, plant/fungal/bacterial
Glyma.20G001400	peroxidase	Haem peroxidase, plant/fungal/bacterial
Glyma.02G085600	HSP70	Heat shock protein 70 family
Glyma.02G184300	NB-ARC	Leucine-rich repeat
Glyma.18G141900	MFS_1	Major facilitator superfamily domain
Glyma.10G212300	Mlo	Mlo-related protein
Glyma.13G006700	NUDIX	NUDIX hydrolase domain
Glyma.14G004300	NUDIX	NUDIX hydrolase domain
Glyma.20G062000	NUDIX	NUDIX hydrolase domain
Glyma.03G096500	2OG-FeII_Oxy	Oxoglutarate/iron-dependent dioxygenase
Glyma.07G124400	2OG-FeII_Oxy	Oxoglutarate/iron-dependent dioxygenase
Glyma.08G169100	2OG-FeII_Oxy	Oxoglutarate/iron-dependent dioxygenase
Glyma.14G058700	2OG-FeII_Oxy	Oxoglutarate/iron-dependent dioxygenase
Glyma.19G194300	PBP	Phosphatidylethanolamine-binding protein PEBP
Glyma.01G010200	PRK	Phosphoribulokinase/uridine kinase
Glyma.09G210900	PRK	Phosphoribulokinase/uridine kinase
Glyma.07G184000	Pkinase/Lectin_LegB	Protein kinase domain
Glyma.17G173000	Pkinase	Protein kinase domain
Glyma.18G219600	Pkinase_Tyr/DUF26	Protein kinase domain
Glyma.13G046200	RuBisCO_small	Ribulose bisphosphate carboxylase small chain, domain
Glyma.10G008500	Response_reg	Signal transduction response regulator, receiver domain
Glyma.08G010400	Auxin_inducible	Small auxin-up RNA
Glyma.06G248700	TIR_2	Toll/interleukin-1 receptor homology (TIR) domain
Glyma.16G137300	TIR_2	Toll/interleukin-1 receptor homology (TIR) domain
Glyma.16G214800	EF-hand_7	Toll/interleukin-1 receptor homology (TIR) domain
Glyma.12G062000	GRAS	Transcription factor GRAS
Glyma.19G214600	zf-C2H2_6	Zinc finger, C2H2

**Table 3 ijms-24-16232-t003:** PI 437654 upregulated genes that were assigned in over-represented GO terms in 10 dpi.

Gene Name	PFAM Description	InterPro Description
Glyma.04G227900	2OG-FeII_Oxy	Oxoglutarate/iron-dependent dioxygenase
Glyma.06G137000	2OG-FeII_Oxy	Oxoglutarate/iron-dependent dioxygenase
Glyma.08G050400	2OG-FeII_Oxy	Oxoglutarate/iron-dependent dioxygenase
Glyma.16G017500	2OG-FeII_Oxy	Oxoglutarate/iron-dependent dioxygenase
Glyma.18G273200	2OG-FeII_Oxy	Oxoglutarate/iron-dependent dioxygenase
Glyma.19G083900	Aa_trans	Amino acid transporter
Glyma.12G015300	ADH_zinc_N	Alcohol dehydrogenase superfamily, zinc-type
Glyma.17G030100	Bet_v_1	Bet v I/Major latex protein/START-like domain
Glyma.17G030300	Bet_v_1	Bet v I/Major latex protein/START-like domain
Glyma.17G030400	Bet_v_1	Bet v I/Major latex protein/START-like domain
Glyma.05G237200	Calmodulin_bind	Calmodulin-binding protein 60
Glyma.07G093900	Calmodulin_bind	Calmodulin-binding protein 60
Glyma.09G182400	Calmodulin_bind	Calmodulin-binding protein 60
Glyma.19G229500	Calmodulin_bind	Calmodulin-binding protein 60
Glyma.01G108200	Cu-oxidase_2	Laccase
Glyma.06G317800	DER1	Derlin
Glyma.03G147700	Dirigent	Plant disease resistance response protein
Glyma.19G151200	Dirigent	Plant disease resistance response protein
Glyma.11G129300	Glyco_hydro_1	Glycoside hydrolase, family 1
Glyma.12G054200	Glyco_hydro_1	Glycoside hydrolase, family 1
Glyma.13G346700	Glyco_hydro_19	Chitin-binding/Glycoside hydrolase
Glyma.05G161300	GST_C	Glutathione S-transferase,/Thioredoxin-like fold
Glyma.08G118700	GST_N	Glutathione S-transferase,/Thioredoxin-like fold
Glyma.08G174900	GST_N	Glutathione S-transferase,/Thioredoxin-like fold
Glyma.14G014700	Hexapep	Serine acetyltransferase, N-terminal/Hexapeptide repeat
Glyma.13G180200	HSF_DNA-bind	Heat shock factor (HSF)-type, DNA-binding
Glyma.08G105400	K-box	Transcription factor, K-box/Transcriptional regulator
Glyma.19G263300	Lipoxygenase	Lipoxygenase
Glyma.16G135200	NB-ARC	Toll/interleukin-1 receptor homology (TIR) domain
Glyma.01G211800	NmrA	NAD(P)-binding domain
Glyma.11G070500	NmrA	NmrA-like domain
Glyma.11G070500	NmrA	NmrA-like domain
Glyma.11G070600	NmrA	NmrA-like domain
Glyma.01G135200	p450	cytochrome P450 (CYP)
Glyma.02G156100	p450	cytochrome P450 (CYP)
Glyma.03G143700	p450	cytochrome P450 (CYP)
Glyma.05G022100	p450	cytochrome P450 (CYP)
Glyma.09G049200	p450	cytochrome P450 (CYP)
Glyma.10G114600	p450	cytochrome P450 (CYP)
Glyma.11G062500	p450	cytochrome P450 (CYP)
Glyma.11G062600	p450	cytochrome P450 (CYP)
Glyma.11G062700	p450	cytochrome P450 (CYP)
Glyma.15G156100	p450	cytochrome P450 (CYP)
Glyma.16G195600	p450	cytochrome P450 (CYP)
Glyma.20G148100	PALP	Cysteine synthase/Tryptophan synthase
Glyma.02G233800	peroxidase	Haem peroxidase, plant/fungal/bacterial
Glyma.03G038600	peroxidase	Haem peroxidase, plant/fungal/bacterial
Glyma.03G038700	peroxidase	Haem peroxidase, plant/fungal/bacterial
Glyma.06G145300	peroxidase	Haem peroxidase, plant/fungal/bacterial
Glyma.09G277900	peroxidase	Haem peroxidase, plant/fungal/bacterial
Glyma.11G161600	peroxidase	Haem peroxidase, plant/fungal/bacterial
Glyma.11G162100	peroxidase	Haem peroxidase, plant/fungal/bacterial
Glyma.18G211000	peroxidase	Haem peroxidase, plant/fungal/bacterial
Glyma.20G169200	peroxidase	Haem peroxidase, plant/fungal/bacterial
Glyma.17G224300	Pkinase/Lectin_legB/RVT_2	Protein kinase domain
Glyma.06G084200	PNP_UDP_1	Nucleoside phosphorylase domain
Glyma.13G113100	Pyr_redox_3	Dimethylaniline monooxygenase, N-oxide-forming
Glyma.17G046600	Pyr_redox_3	Flavin monooxygenase FMO
Glyma.17G046500	Redoxin	Redoxin/Thioredoxin-like fold
Glyma.05G204600	Thaumatin	Thaumatin
Glyma.10G251500	Thi4	Thiazole biosynthetic enzyme Thi4 family
Glyma.08G277000	Transketolase_C	Transketolase-like, pyrimidine-binding domain
Glyma.02G029400	zf-C2H2_6	Zinc finger, C2H2

**Table 4 ijms-24-16232-t004:** Sets of genes in over-represented GO terms of PI 437654 that were unique DEGs or validated by PI 548402 and/or PI 88788 at 5 dpi.

Genes	Functional Category	PI 437654	PI 548402	PI 88788
Glyma.08G010400	Auxin responsive protein	X		
Glyma.03G232400	Calmodulin-binding protein 60	X		
Glyma.19G229400	Calmodulin-binding protein 60	X		
Glyma.03G242100	Copine, C2 domain	X		
Glyma.15G250800	FMN-dependent dehydrogenase	X		
Glyma.11G111400	Fructose-bisphosphate aldolase	X		
Glyma.12G037400	Fructose-bisphosphate aldolase	X		
Glyma.01G010200	Phosphoribulokinase	X		
Glyma.09G210900	Phosphoribulokinase	X		
Glyma.02G085600	HSP70, C1 domain	X		
Glyma.16G165500	Light-harvesting complexes or Chlorophyll A-B-binding protein	X		
Glyma.13G046200	RuBisCO	X		
Glyma.09G087700	Photosystem I psaG/psaK	X		
Glyma.18G141900	Major facilitator family	X		
Glyma.12G009200	Methyltransferase domain,	X		
Glyma.08G324300	Glycerophosphoryl diester phosphodiesterase family	X		
Glyma.19G194300	Phosphatidylethanolamine-binding protein	X		
Glyma.03G194700	Phosphatidylethanolamine-binding, conserved site	X		
Glyma.15G052700	Plant peroxidase	X		
Glyma.10G251500	Thiazole biosynthetic enzyme Thi4 family	X		
Glyma.12G062000	Transcription factor GRAS	X		
Glyma.19G214600	Zinc finger protein GIS3/ZFP5/ZFP6	X		
Glyma.14G058700	2OG-Fe(II) oxygenase superfamily	X	X	
Glyma.03G096500	2OG-Fe(II) oxygenase superfamily	X	X	
Glyma.07G124400	2OG-Fe(II) oxygenase superfamily	X	X	
Glyma.08G169100	2OG-Fe(II) oxygenase superfamily	X	X	
Glyma.19G239500	C2 domain/Copine	X	X	
Glyma.04G035600	cytochrome P450 (CYP)	X	X	
Glyma.09G144300	cytochrome P450 (CYP)	X	X	
Glyma.17G054600	Cytokinin dehydrogenase 1, FAD and cytokinin-binding	X	X	
Glyma.20G087000	ethylene receptor	X	X	
Glyma.15G134200	FAD-binding domain/Berberine and berberine-like	X	X	
Glyma.15G134300	FAD-binding domain/Berberine and berberine-like	X	X	
Glyma.14G004300	NUDIX domain	X	X	
Glyma.03G038300	Peroxidase	X	X	
Glyma.19G011800	Peroxidase/oxygenase	X	X	
Glyma.10G008500	Response regulator receiver domain	X	X	
Glyma.06G248700	TIR domain	X	X	
Glyma.02G028000	Peptidoglycan binding domain, peptidase	X		X
Glyma.03G162400	AP2 domain	X	X	X
Glyma.03G162700	AP2 domain	X	X	X
Glyma.05G186700	AP2 domain	X	X	X
Glyma.10G186800	AP2 domain	X	X	X
Glyma.13G122500	AP2 domain	X	X	X
Glyma.13G123100	AP2 domain	X	X	X
Glyma.19G163700	AP2 domain	X	X	X
Glyma.19G163900	AP2 domain	X	X	X
Glyma.01G130500	Peroxidase	X	X	X
Glyma.03G038500	Peroxidase	X	X	X
Glyma.09G023000	Peroxidase	X	X	X
Glyma.09G057100	Peroxidase	X	X	X
Glyma.20G001400	Peroxidase	X	X	X
Glyma.07G184000	Protein kinase domain	X	X	X
Glyma.17G173000	Protein kinase domain	X	X	X
Glyma.08G018300	WRKY DNA -binding domain	X	X	X
Glyma.19G229500	Calmodulin binding protein-like	X	X	X
Glyma.02G184300	Leucine rich repeat	X	X	X
Glyma.20G036100	Ribonuclease T2 family	X	X	X
Glyma.12G053900	Glycosyl hydrolase family 1	X	X	X

**Table 5 ijms-24-16232-t005:** Sets of the genes in over-represented GO terms of PI 437654 that were unique DEGs or validated by PI 548402 and/or PI 88788 at 10 dpi.

Gene ID	Functional Category	PI 437654	PI 548402	PI 88788
Glyma.07G153800	Ammonium_transp	X		
Glyma.10G168100	Ammonium_transp	X		
Glyma.14G014700	Hexapeptide repeat	X		
Glyma.20G148100	Tryptophan synthase	X		
Glyma.10G251500	Thiazole biosynthetic enzyme Thi4 family	X	X	
Glyma.11G129300	Glycoside hydrolase, family 1	X	X	
Glyma.05G237200	Calmodulin_bind protein	X		X
Glyma.07G093900	Calmodulin_bind protein	X		X
Glyma.19G229500	Calmodulin_bind protein	X		X
Glyma.19G151200	Dirigent	X		X
Glyma.13G180200	Heat shock factor (HSF)	X		X
Glyma.19G263300	Lipoxygenase	X		X
Glyma.09G277900	peroxidase	X		X
Glyma.20G169200	peroxidase	X		X
Glyma.17G224300	Pkinase	X		X
Glyma.17G046500	Thioredoxin-like	X		X
Glyma.05G204600	Thaumatin	X		X
Glyma.04G223300	WRKY	X		X
Glyma.13G267600	WRKY	X		X
Glyma.17G222300	WRKY	X		X
Glyma.04G227900	2OG-Fe(II) oxygenase superfamily	X	X	X
Glyma.18G273200	2OG-Fe(II) oxygenase superfamily	X	X	X
Glyma.14G038400	Ankyrin repeat-containing domain	X	X	X
Glyma.17G030100	Bet_v_1	X	X	X
Glyma.09G182400	Calmodulin_bind protein	X	X	X
Glyma.01G108200	Laccase	X	X	X
Glyma.06G317800	Derlin	X	X	X
Glyma.03G024200	Glycoside hydrolase family	X	X	X
Glyma.07G103700	Late embryogenesis abundant protein	X	X	X
Glyma.03G116300	MATH	X	X	X
Glyma.02G054200	Methyltransf_7	X	X	X
Glyma.18G238800	Methyltransf_7	X	X	X
Glyma.13G035900	Pkinase	X	X	X
Glyma.11G207000	Pkinase_Tyr/DUF26	X	X	X
Glyma.13G113100	Pyr_redox_3	X	X	X
Glyma.11G210300	UbiA	X	X	X
Glyma.13G267500	WRKY	X	X	X
Glyma.14G103100	WRKY	X	X	X
Glyma.18G213200	WRKY	X	X	X
Glyma.15G219400	zf-MYND	X	X	X
Glyma.07G098600	putative	X	X	X
Glyma.10G161500	putative	X	X	X
Glyma.07G098700	putative	X	X	X
Glyma.08G085700	putative	X	X	X

**Table 6 ijms-24-16232-t006:** Differentially expressed SCN putative effector genes in different soybean lines: PI 437654, PI 548402, PI 88788, Lee 74.

Identified Effector		Probability of Predicted Signals and Localizations			Significantly Upregulated				Reported Effectors
Genome Draft Format-ID	Pseudomolecule Assembly-ID	Signal Peptide	Extra-Cellular	Chr	PI 437654	PI 548402	PI 88788	Lee 74	
Hetgly.G000000386	Hetgly10240	0.9556	0.8421	2	X				NO
Hetgly.G000002041	Hetgly00243	0.8547	0.5982	5	X				NO
Hetgly.G000003725	Hetgly01606	0.9997	0.9750	5			X	X	NO
Hetgly.G000003742	Hetgly01620	0.5363	0.7217	5		X	X	X	NO
Hetgly.G000005495	Hetgly01486	0.9995	0.9838	5	X				NO
Hetgly.G000005767	Hetgly10721	0.9997	0.5519	2	X				NO
Hetgly.G000005859	Hetgly07384	0.9996	0.9257	1		X			NO
Hetgly.G000006034	Hetgly15160	0.9996	0.7696	6	X				NO
Hetgly.G000006271	Hetgly13266	0.9951	0.9072	4	X				YES ^1,2^
Hetgly.G000007737	Hetgly14373	0.9997	0.9553	6	X				YES ^1^
Hetgly.G000008328	Hetgly08786	0.9998	0.9278	1	X				YES ^1^
Hetgly.G000008629	Hetgly10435	0.8469	0.7313	2			X		NO
Hetgly.G000008756	Hetgly04908	0.9998	0.9513	3		X	X	X	NO
Hetgly.G000008760	Hetgly04892	0.9997	0.9781	3	X				YES ^1^
Hetgly.G000009320	Hetgly14722	0.8276	0.9354	6		X			NO
Hetgly.G000009584	Hetgly20383	0.9997	0.9577	9		X	X	X	NO
Hetgly.G000009600	Hetgly20345	0.9997	0.9135	9			X	X	NO
Hetgly.G000009601	Hetgly20346	0.9992	0.8075	9			X	X	NO
Hetgly.G000010445	Hetgly05887	0.9648	0.9543	3		X	X	X	NO
Hetgly.G000010619	Hetgly08167	0.9997	0.9611	1	X				YES ^1^
Hetgly.G000011018	Hetgly00611	0.9971	0.9794	5	X				NO
Hetgly.G000011037	Hetgly21117	0.9997	0.9685	8		X	X	X	NO
Hetgly.G000011113	Hetgly05749	0.9998	0.9162	3				X	NO
Hetgly.G000011117	Hetgly05752	0.9998	0.9662	3		X			NO
Hetgly.G000011166	Hetgly05791	0.9997	0.6696	3	X				NO
Hetgly.G000011538	Hetgly06927	0.7568	0.5704	1	X				NO
Hetgly.G000011604	Hetgly19536	0.6212	0.9683	9		X	X		NO
Hetgly.G000011607	Hetgly19535	0.9996	0.9769	9		X	X	X	NO
Hetgly.G000012678	Hetgly20828	0.9996	0.5010	8	X				NO
Hetgly.G000013560	Hetgly06640	0.9998	0.7309	1			X	X	YES ^1^
Hetgly.G000014327	Hetgly13555	0.9998	0.9694	4	X				YES ^1^
Hetgly.G000014444	Hetgly02677	0.9997	0.9754	5		X	X	X	NO
Hetgly.G000014464	Hetgly02651	0.9997	0.9599	5	X				NO
Hetgly.G000014527	Hetgly09638	0.8601	0.8579	2	X				NO
Hetgly.G000015939	Hetgly07801	0.9998	0.9350	1				X	NO
Hetgly.G000016234	Hetgly16974	0.9808	0.9515	7				X	NO
Hetgly.G000016328	Hetgly13997	0.9994	0.9480	4			X	X	YES ^1^
Hetgly.G000016675	Hetgly05522	0.9997	0.9099	3	X				NO
Hetgly.G000016740	Hetgly11688	0.9997	0.9764	2		X		X	NO
Hetgly.G000016899	Hetgly14389	0.8984	0.6139	6	X				NO
Hetgly.G000017118	Hetgly11576	0.9998	0.9692	2	X				YES ^1^
Hetgly.G000017651	Hetgly09041	0.9963	0.8390	2				X	NO
Hetgly.G000017808	Hetgly20463	0.9998	0.8331	9			X	X	NO
Hetgly.G000018759	Hetgly06909	0.9997	0.9492	1				X	NO
Hetgly.G000018760	Hetgly06908	0.9997	0.9683	1		X			YES ^1^
Hetgly.G000018896	Hetgly03920	0.9998	0.9405	3		X			NO
Hetgly.G000019218	Hetgly20700	0.9998	0.9576	8			X	X	NO
Hetgly.G000021894	Hetgly08087	09985	0.7620	1	X				YES ^1^
Hetgly.G000022800	Hetgly08816	0.9998	0.9282	1		X	X	X	YES ^1^
Hetgly.G000023464	Hetgly12039	0.9998	0.8796	4	X				NO
Hetgly.G000028400	Hetgly03197	0.9997	0.9618	3		X	X	X	NO

^1^: Masonbrink R. et al. [25]; ^2^: Teufel F. et al. [26].

## Data Availability

The raw reads mRNA-seq reads can be downloaded at the NCBI SRA. The RNA-seq reads are deposited to the NCBI SRA under BioProject PRJNA972659 (https://www.ncbi.nlm.nih.gov/sra/PRJNA972659, accessed on 6 June 2024).

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
