# Peer review of "Soybean–SCN Battle: Novel Insight into Soybean’s Defense Strategies against Heterodera glycines"

_ijms, 2023, doi:10.3390/ijms242216232_

Round 1
Reviewer 1 Report
Comments and Suggestions for Authors
In this manuscript, the authors use RNA-seq to elucidate the resistance mechanisms for SCN resistance in several accessions of soybean, as well as SCN genes. They find differentially expressed genes in resistant and susceptible accessions, allowing better understanding of the biology of the system.
Overall the research is clear, integrates information between molecular genetics and SCN resistance, and has some interesting results concerning differences in resistance mechanisms. There are no major flaws, but there are a few minor things which could be better explained.
As a minor typographical note, please ensure that binomial nomenclature is correct and consistent, as it is variable in figures and legends.
As another minor typographical note, figure four and its legend could be clearer and more correctly labeled.
More discussion could be spent on the rhg1/Rgh4 gene functions. Are they constitutively more highly expressed, do they function through a mechanism not involved with their differential expression, or ae they irrelevant for this SCN race?
Concerning line 614, is there evidence that SCN (putative) effector modulation plays a role in overcoming resistant hosts? They may be different but I don’t see evidence they are adaptive.
Concerning line 711, did you do any, for instance, cytological work to demonstrate that the cell walls are actually altered? This seems like it would the case given the RNA data, but the relationship between RNA and function is often nuanced.
Concerning line 722, confirming this with genomic data would be very interesting, but likely beyond the scope of this manuscript.
Concerning line 724, I don’t see which data supports that. I don’t follow how different hosts resulting in different gene expression implies mutation. A check on how inbred the SCN lines were would also be nice, but might not be possible.
Author Response
Dear Reviewer,
First and foremost, we would like to express our sincere appreciation to you for generously dedicating your time and providing us with invaluable comments. We have diligently incorporated the majority of these comments into our manuscript, ensuring that your insights have significantly contributed to its improvement. For the few comments that we chose not to incorporate, we have thoughtfully considered the reasons behind this decision. Please find our responses to your comments in the attached file.
Sincerely,
Milad Eskandari

Reviewer 2 Report
Comments and Suggestions for Authors
Some of subtopic sentences in the results should be revised.
Author Response

(The authors gave the same response as above.)

Reviewer 3 Report
Comments and Suggestions for Authors
If the editor decides that the publication does not infringe copyright, the article seems to be within the scope of journal. However it needs several corrections to be more acceptable for publication.
Title of fig. 3 It should be „H. glycines” instead of „H. Glycine”.
In table 2, 3, and 4: it should be „calmodulin-binding protein 60” instead of „CALMODULIN-BINDING PROTEIN60” (Shen, L., He, J., Yang, X. Genome-wide identification of calmodulin-binding protein 60 gene family and function of SmCBP60A1 in eggplant response to salt stress. Scientia Horticulturae, 2023, 322, 112448).
Line 278: It should be „collagen” instead of „Collagen”.
Table 4 It should be written „RuBisCO”, because it is the enzyme ribulose bisphosphate carboxylase/oxygenase (EC.4.1.1.39; RuBisCO), a protein that plays a vital role in photosynthesis. (Grácio, M., Oliveira, S., Lima, A., Ferreira, R. B. RuBisCO as a protein source for potential food applications: a review. Food Chemistry, 2023, 135993.).
Table 5: If the column is named "Functional Category", the names of proteins should consistently be included as in the previous table, e.g. it should be „2OG-Fe(II) oxygenase superfamily” instead of „2OG-FeII_Oxy”.
The text on lines 355-363 is written in the wrong font.
Lines 421, 422: It should be „Ca2+” instead of „Ca2+”. Please correct it
Line 461: It should be „G. soja” instead of „G. soja”.
Line 471: The notation for all radicals is incorrect. Please correct it.
Line 484” it should be „glutathione” instead of „Glutathione”.
Lines 491, 719: it should be „2-oxoglutarate/Fe(II)-dependent oxygenase” instead of „2OG-Fe(II) oxygenase”. Please correct it.
Line 492, 720: it should be „cytochrome P450s” (CYP)” instead of „Cytochrome P450 (P450)”, because it is a superfamily of enzymes – the largest enzyme family involved in NADPH- and/or O2-dependent hydroxylation reactions across all the domains of life. See: Pandian, B. A., Sathishraj, R., Djanaguiraman, M., Prasad, P. V., Jugulam, M. Role of cytochrome P450 enzymes in plant stress response. Antioxidants, 2020, 9(5), 454.
Line 524: It should be „laccase” instead of „Laccase”.
Line 713: it should be „H. glycines” instead of „H. glycines”. Please correct it.
References 1 and 2 are very old. Please use newer sources for this information.
Author Response

(The authors gave the same response as above.)

Reviewer 4 Report
Comments and Suggestions for Authors
I have enjoyed reading the manuscript by Torabi et al., describing transcriptional interactions between and withing different soybean cultivars and nematodes. The results provide certainly valuable information.
Main question addressed by the research: Global gene expression changes in SCN-resistant Plant Introduction (PI) 437654, 548402, and 88788 as well as a susceptible line (Lee74) under exposure to SCN HG type 1.2.5.7.
I do believe the results from these manuscripts are original transcriptome data, and provides additional data into the field of soybean/SCN interactions. The data by Torah et al. provide transcriptome results of not only soybean response to SCN, but also SCN response to soybean.
The transcriptome data is chiefly considered as the reference data, but does not really aim at deciphering the intricate molecular mechanisms. Hence, considering a limited budget and a complexity of real cellular/physiological mechanisms, it’s usually unreasonable to demand all physiological/cellular controls for transcriptome study. I am satisfied with the experiments since it is a part of efforts to enrich the reference data for those who look for the starting point to investigate the actual mechanism/resources to study soybean defense responses against SCN.
This study would provide reference data for the future study. Again, I do believe that is what the transcriptome study can suppose to achieve.
A concern is the lack of details in the manuscript – should be revised before being published. For instance;
Line 37: remove a space between of and $.
Line 42; remove ‘s’ from defences.
Line 46 and 51: PTI is explained twice.
Line 59: remove (NB-LRR) since it used only once.
Throughout manuscript, there are such many minor errors in the repeated abbreviations, the italicization of genes, empty spaces and so on.
1. In addition, table 1 can go to the supplemental data, as it’s from the other study.
2. Line 122: what does ‘FI’ stand for and mean?
3. Figure 1. P I188788 should be PI 88788. Why A is missing Lee 74?
4. Line 141: remove a space between SE and (b).
5. Figure 2: why Williams?
6. Line 171: again Lee74 should be Lee 74.
7. Throughout Figures: the most of all, need to add proper statics, and arrange the order of hosts in a consistent manner (e.g., Lee, 437654, 548402, 88788) – and check a space between Lee/PI and numbers.
8. 8 Line 260: why explain SCN again there?
9. Line 278: Collagen to collagen.
10. Line 325: Rewite the subtitle – cannot understand it as it is.
11. Line 328: remove a space between the and behavior.
12. Figure 6: a number 86 under SCN|PI 88788 is a half disappeared.
Not only those mentioned above but authors should carefully revised those minor errors.
Author Response

(The authors gave the same response as above.)

Round 2
Reviewer 3 Report
Comments and Suggestions for Authors
The authors revised the manuscript and I accept it in present form.